# Microbiological and Molecular Investigation of Antimicrobial Resistance in *Staphylococcus aureus* Isolates from Western Romanian Dairy Farms: An Epidemiological Approach

**DOI:** 10.3390/ani14152266

**Published:** 2024-08-04

**Authors:** Ioan Hutu, Bianca Cornelia Lungu, Ioana Irina Spataru, Iuliu Torda, Tiberiu Iancu, Paul Andrew Barrow, Calin Mircu

**Affiliations:** 1“Horia Cernescu” Research Unit, Faculty of Veterinary Medicine, University of Life Sciences “Regele Mihai I”, Calea Aradului 119, 300645 Timisoara, Romania; ioanhutu@usvt.ro (I.H.); bianca.lungu@fmvt.ro (B.C.L.); ioana.spataru@usvt.ro (I.I.S.); iuliu.torda@usvt.ro (I.T.); calinmircu@usvt.ro (C.M.); 2Faculty of Agricultural Management, University of Life Sciences “Regele Mihai I”, Calea Aradului 119, 300645 Timisoara, Romania; 3School of Veterinary Medicine, University of Surrey, Daphne Jackson Rd., Guildford GU2 7AL, UK

**Keywords:** *Staphylococcus aureus*, mastitis, antimicrobial resistance, penetrance, dairy farms

## Abstract

**Simple Summary:**

Antimicrobial therapy is the most frequently used medical intervention for bovine mastitis in the dairy industry. In this study, we aimed to monitor the extent of the antimicrobial resistance problem in *Staphylococcus aureus*. The multiple antibiotic resistance was calculated for bovine and human strains, using phenotypic and genotypic methods. It is a cause for concern that values were higher than acceptable limits for almost all antibiotics currently used in the dairy industry. Genotypic analysis was used for assessing the degree of penetrance of the resistance genes in *Staphylococcus*. Given that penetrance in *S. aureus* was strongly positively correlated with the multi-antibiotic resistance index (MARI), it may be possible to use the same limit value for management decision purposes. In fact, values of penetrance over 0.20, along with MARI values over 0.40, raise questions over the validity of current antimicrobial treatment programs/strategies, which the study suggests ought to be changed.

**Abstract:**

Antimicrobial therapy is the most frequently used medical intervention for bovine mastitis in the dairy industry. This study aims to monitor the extent of the antimicrobial resistance (AMR) problem in *Staphylococcus aureus* in the dairy industry in Western Romania. Twenty farms were selected by random sampling in a transverse epidemiological study conducted across four counties in Western Romania and divided into livestock units. This study assessed the association between the resistance genes to phenotypic expression of resistance and susceptibility. Isolates of *S. aureus* were identified and q-PCR reactions were used to detect antibiotic resistance genes. One hundred and fifty bovine and 20 human samples were positive for *S. aureus*. Twenty five percent of bovine isolates (30/120) and none(0/30) of the human isolates were methicillin-resistant *S. aureus* (MRSA). All isolates were susceptible to fosfomycin, ciprofloxacin, netilmicin, and resistant to ampicillin and penicillin. *S. aureus* isolates regarded as phenotypically resistant (R) were influenced by the origin of the samples (human versus bovine, χ2 = 36.510, *p* = 0.013), whether they were methicillin-resistant *S. aureus* (χ2 = 108.891, *p* < 0.000), the county (χ2 = 103.282, *p* < 0.000) and farm of isolation (χ2 = 740.841, *p* < 0.000), but not by the size of the farm (χ2 = 65.036, *p* = 0.306). The multiple antibiotic resistance index was calculated for each sample as the number regarded as phenotypically resistant (R)/total antibiotics tested (MARI = 0.590 ± 0.023) was significantly higher (*p* < 0.000) inmethicillin-resistant *S. aureus* (0.898 ± 0.019) than non-methicillin-resistant *S. aureus* (0.524 ± 0.024) isolates. For the antibiotics tested, the total penetrance (P%) of the resistance genes was 59%, 83% for *blaZ*, 56% for *cfr*, 50% for *erm(B)*, 53% for *erm(C)*, 57% for *mecA* and 32% for *tet(K)*. Penetrance can be used as a parameter for guidance towards a more accurate targeting of chemotherapy. P% in *S. aureus* was strongly positively correlated with the multiple antibiotic resistance index (r = +0.878, *p* < 0.000) with the potential to use the same limit value as an antibiotic management decision criterion. Considering cow mastitis, the penetrance value combined with the multiple antibiotic resistance index suggests that penetrance could serve as a useful parameter for more precise targeting of chemotherapy for *S. aureus*.

## 1. Introduction

Bovine mastitis is a multifactorial inflammatory disease involving a combination of pathogen, host genetic and environmental factors [1,2]. It is the most common disease which leads to economic loss in the dairy industry, as a result of reduced yield, poor milk quality [3], discarded milk, increased replacement costs, and the cost of treatment and other veterinary services [4]. The losses in the United States are estimated at USD 2 billion p.a., in the United Kingdom, GBP 300 million p.a. and in the Netherlands, the estimated cost varies from EUR 114 to EUR 182/cow per annum [5,6,7] with milk production losses and culling representing 11% to 18% of the gross margin per cow per year [8]. The mammary tissue damage leading to decreased milk production accounts for 70% of the total losses [9]. Bacterial intramammary inflammation is an infectious or environmental disease depending on the pathogen involved [10,11]. Whereas Gram-negative bacteria are the main cause of environmental infections frequently associated with winter housing and calving, contagious mastitis involves bacteria transmissible between cattle, especially in the milking parlor [12,13]. The predominant contagious bacterial pathogens are *Staphylococcus aureus* and *Streptococcus agalactiae*, which, with less frequently encountered species, such as *Mycoplasma bovis* and *Corynebacterium ulcerans*, live on the cow’s udder and teat skin, colonizing and infecting the teat canal [14].

In addition to the substantial milk production losses associated with mastitis, the disease has serious zoonotic potential with *S. aureus* as a frequent contaminant of foodstuffs, and possibly representing the main food-borne pathogen causing health problems in both humans and animals [15]. In humans, *S. aureus* colonizes the nasal mucosa and skin in up to 50% of the healthy population, and it can also be responsible for blood infections [16]. Control of infectious bovine mastitis is best managed by breaking the infection cycle, applying antiseptic teat-dipping immediately after milking [17,18,19]. This has been largely successful in many countries and clearly would have positive outcomes in Romania in terms of both reduced levels of mastitis but also reduced zoonotic transmission [20,21,22]. Vaccines are not very effective against Gram-positive pathogens [23] and novel approaches, including lytic bacteriophages, are being considered [24]. Chemotherapy continues to be used against Gram-positive infections in many countries and, in countries where environmental mastitis is more prevalent, treatment of Gram-negative infections inevitably leads to exposure of Gram-positive bacteria to the antimicrobials used.

Resistance to penicillin and other antibiotics is widespread globally [25,26,27,28,29]. Levels of antimicrobial resistance (AMR) vary considerably between countries, even within Europe [30], most probably dependent on the strategy used to control infection [31,32]. Multi-drug-resistant strains are increasingly isolated, posing a huge problem both for livestock and humans [33,34,35,36,37,38]. Treatment may be complicated by biofilm formation by staphylococci in the udder parenchyma by making the bacteria more intractable to antimicrobials [39]. Antibiotic-resistant bacteria are excreted by humans and animals via feces, body fluids and skin, leading to contamination of the environment where gene exchange can also occur with environmental bacteria. Animals and man may thus become exposed to antibiotic resistant bacteria either by direct physical contact or indirect contact via the environment and fomites [40]. If withdrawal periods are ignored after chemotherapy drug residues in milk may also impact community public health [41]. Spread and dissemination of antimicrobial resistant strains of *S. aureus* may occur by transmission between individual hosts of a single clone or by transmission of the genetic determinants between AMR and antibiotic susceptible strains or other *Staphylococcus* species, via plasmids and bacteriophage activity [42,43,44]. These are processes clearly influenced by antibiotic use in human and veterinary medicine [45]. In response to the concern within international institutions, including FAO and OIE [46] over the endemicity of AMR in many veterinary bacterial pathogens, antibiotic usage has declined in the last few years [47]. Overall aggregated sales for the 25 countries participating in the European Surveillance of Veterinary Antimicrobial Consumption (ESVAC) project reached the lowest ever reported values in 2022, having declined 53.0% since 2011 (from 161.2 mg/PCU to 75.8 mg/PCU). Compared to other European countries, in 2022, the sales in Romania were (see Figure 1) lower than 2021 with 17.3% (59.0 vs. 48.8 mg/PCU). The ranking for the highest-selling antibiotics at 21.8%, 17.8% and 15.3% of total sales consisted of tetracyclines, penicillins and macrolides [48]. The antimicrobials most used for mastitis treatment in the Romanian dairy farms sampled are currently tetracycline (Mastijet Fort^®^), ampicillin (Mamifort^®^) and amoxicillin (Clamox LC^®^).

The aim of the study was to isolate *S. aureus* strains with comprehensive phenotypic and genotypic characterization, including drug resistance, in order to better understand the extent of the AMR problem in the dairy sector in West Romania. To do this we characterized antibiotic resistance phenotypes and genotypes in *S. aureus* isolates by a transverse epidemiological study in dairy farms involving milking cattle and humans in direct contact with the animals. We investigated antibiotic resistance genes and assessed the association and correlation between specific resistance genes and the phenotypic expression of resistance and susceptibility—in order to calculate the multiple antibiotic resistance index (MARI) and the penetrance as a potential indicator for management and clinical usage of antimicrobials.

## 2. Materials and Methods

### 2.1. Description of Sampling

The study took place in the four counties, Bihor, Arad, Timiș and Caras-Severin in the Western region of Romania, involving all dairy farms registered in Official Control of Milk production, during a one and half year period (January 2022 to July 2023). The randomly selected farms for the transverse epidemiological trial were stratified by county (5 farms for each 4 county = 20 farms) and a minimum of one farm selected from each of the four classes of Livestock Units (1–25, 26–50, 51–100, and over 100 LU/farm).Based on previous studies [13], a point prevalence of 40% was assumed for calculating of the sampling volume. A minimum of six animals (5.86) per farms were proposed to be sampled, using the formula n = log β/log p, where β is the probability of committing a type II error (set at ≤0.05 for this study) and p represents the proportion of uninfected animals. In the selected farms, the cows subjected to the study were initially selected based on the official milk analysis report from the farm. All cows with a somatic cell count greater than 200,000 cells/mL milk in a previous official test (individual samples processed in less than 14 days before) were tested with the California Mastitis Test (CMT).Samples were collected from the cows identified positive by CMT (with scores ranging from weak to distinctly or heavily positive), in order to carry out a bacteriological examination: an average was collected 16.25 + 1.50 samples pereach selected farm [7,13]. The milk samples were collected (Figure 2) from three breeds: Holstein cows with 1.74 ± 0.45 lactations, producing 27.55 ± 1.80 kg of milk per day, Romanian Simmental cows with 2.39 ± 0.72 lactations, producing 21.03 ± 2.76 kg of milk per day, and Romanian Brown of Maramuresh cows with 2.21 ± 0.42 lactations, producing 22.45 ± 1.87 kg of milk per day.

Samples were also taken on eSwabs from both nostrils of the milkers and employees in 60% of the farms (12 farms), respecting the voluntary nature of sample provision and GDPR. All samples were transported in eSwabs and arrived at the microbiology laboratory within 12–24 h after collection.

### 2.2. Samples and Staphylococcus aureus Isolation

A total of 325 CMT-positive milk samples were collected from cows, along with 30 samples from human nasal mucosa. These samples were gathered using ESwab™ (COPAN Diagnostics, Murrieta, CA, USA) and transported to the laboratory in a protective pack. For presumptive identification of cultures from the swab content, Columbia Blood agar (BIOLAB, Budapest, Hungary) was used for inoculation, streaking out with incubation at 37 °C for 24 h. Presumptive *Staphylococcus* colonies were picked and repurified on Columbia Blood agar and identified by colonial morphology. The colonies were chosen by their round form and white or golden appearance for confirmation and identification with the Micro Scan System. A single purified colony from each sample was transferred to 0.5 mL of nutrient broth (BIOLAB, Budapest, Hungary) and stored at −80°C for potential future q-PCR analysis following confirmation.

### 2.3. Confirmation of Staphylococcus aureus and Detection of Antimicrobial Resistance

The Micro Scan Walk Away 40SI System(Dade Behring, West Sacramento, CA, USA) was used according to the technical instructions, for confirmation and identification of *S. aureus* strains and identification, using MIC, minimum inhibitory concentration from the Positive Breakpoint Combo Panel Type 29 (PBC-29, Beckman Coulter, Inc, Brea, California, USA ) of antimicrobial resistance and/or susceptibility. For Linezolid and Oxacillin, the MIC was increased in according toCLSI—2015 methods MIC tests. The automated system’s barcode allocation categorized the samples as S (antibiotic-susceptible), I (intermediate), and R (antibiotic-resistant).The outputs for cefoxitin screening were positive (POS) for resistant (R) and negative (NEG) for susceptible (S) strains. In the case of ampicillin and penicillin, there were some βlac (beta lactamase)-positive outputs which were also considered to be resistant (R). In this case intermediate (I), resistant (R), POS and βlac outputs were all grouped together as resistant.

### 2.4. DNA Extraction and Detection of Resistance Genes viaq-PCR

To extract genomic and plasmid DNA, the DNeasy Blood and Tissue Kit (Qiagen, Hilden, Germany) was utilized according to the manufacturer’s instructions. DNA concentration was estimated using the Nano Quant Plate™ (Tecan Trading AG, Männedorf, Switzerland) set at 260 nm/280 nm absorbance. The conditions of the reaction procedure were as follows: pre-denaturation at 95 °C for 10 min, denaturation at 95 °C for 30 s, annealing for 1 min at Tm°C (Tm/°C of all genes are shown in Table 1), extension at 72 °C for 1 min for 40 cycles. The optical configuration used for SYBR Green was between 492 nm and 516 nm. For the quantitative PCR, a mixture was prepared consisting of 25 ng of bacterial DNA, water, 12.5 µL of SYBR™ Green Master Mix (Thermo Fisher Scientific, Waltham, MA, USA), and 1 µL each of the forward (FW) and reverse (RV) primers for the *blaZ*, *erm(B)*, *erm(C)*, *cfr*, *mecA*, or *tet(K)* genes (Metabion International AG, Planegg, Germany).For each primer set, a master mix was made, including SYBR Green, FW and RV primers (Table 1). To achieve a total reaction volume of 25 µL, the volume of water added was adjusted based on the DNA concentration. The antibiotics that were used were those contained in the PBC-29 panel and most frequently administered in the dairy industry. For identifying the resistance genes through qPCR, the Agilent Technologies Stragene Mx3005P (Agilent Technologies Division, Model nr. 401513, Heidelberg, Germany) was used. A negative reaction was assigned to the samples where amplification started after the 40th cycle of the annealing step. The cut-off for expressing resistance was accepted for the cases with more than 12 cycles (Ct threshold cycles) and fewer than 40 cycles. We used the *Staphylococcus aureus* strain ATCC33591 and the culture collections from County Hospital of Timis as a positive control, and DNA-free water (Qiagen, Hilden, Germany) as a negative control. The q-PCR results for isolates with a resistant phenotype (R and I classes according to Micro Scan’s outputs) were categorized into RG+ (presence of resistance genes) or RG− (absence of resistance genes). Phenotypically susceptible isolates were labelled as SG+ (gene present) or SG− (gene absent).

### 2.5. Phenotypic and Genotypic Resistance

Due to the difference in resistance levels in AST (antibiotic susceptibility testing) registered between strains, the MARI was required to be measured, reflecting the prevalence of resistance or susceptibility [55]. Penetrance, defined as the proportion of bacteria phenotypically resistant due to the presence of resistance genes, was calculated following methods described in a previous study [56]. It was determined by dividing the number of individuals exhibiting the resistance phenotype (R + I categories from Micro Scan outputs) by the total number of individuals with the resistant genotype (RG+ class, determined through q-PCR analysis). Penetrance (P%) was expressed as a percentage [57,58], as elaborated further [56].
$$\mathrm{Penetrance}\;\left(P\%\right)=\frac{\left(\mathrm{RG}+\right)}{\left(\mathrm{RG}+\right)+\left(\mathrm{SG}+\right)}\times \;100$$

The diagnostic odds ratio (DOR) of positive phenotypic resistance [59] was employed to assess the association with genetic resistance indicated by phenotypic antimicrobial susceptibility testing (AST).
$$\mathrm{DOR}=\frac{\left(\mathrm{RG}+\right)\times \left(\mathrm{SG}-\right)}{\left(\mathrm{RG}-\right)\times \left(\mathrm{SG}+\right)}$$

The classifications were defined as follows:(RG+): Phenotypically resistant and possessing the resistance gene.(RG−): Phenotypically resistant but lacking the resistance gene.(SG+): Phenotypically susceptible but with the resistance gene.(SG−): Phenotypically susceptible and without the resistance gene.

As suggested by *Krumperman* in 1983 [60], the multiple antibiotic resistance index (MARI) for individual isolates was calculated and interpreted using the following formula:MAR index = a/b,
where a represents the number of antibiotics to which the isolate exhibited resistance, and b denotes the total number of antibiotics tested against the isolates.

### 2.6. Statistical Analysis

The statistical tests used for interval or continuous variables, included the paired *t*-test, Wilcoxon signed-rank test (non-parametric), Kruskal–Wallis test, Pearson’s correlation and linear regression. Frequencies were analyzed using the Pearson’s chi-squared test. These statistical analyses were performed using SPSS Statistics for Windows, Version 17.0 (SPSS Inc., Chicago, IL, USA). Statistical significance was considered at *p* values of <0.05.

## 3. Results

### 3.1. Epidemiological Results

Of the 325 samples from cows with mastitis and positive to the California Mastitis Test (CMT), 150 (46.1%) were confirmed as *S. aureus*. Of 30 samples from humans, 20 (66.6%) were confirmed as *S. aureus*. There was no association between the frequency with which *S. aureus* was isolated from bovine or human samples (χ2 = 15.867, *p* = 0.667) and no association with farm (χ2 = 15.867, *p* = 0.667) or county (χ2 = 1.989, *p* = 0.575) of origin (Figure 3). At the farm level (20 randomized epidemiological units), between six and nine *S. aureus* isolates were recovered from the animals and between zero and two from human attendants, and the rate of isolation in the two hosts was not statistically significant (χ2 = 15.867, *p* = 0.667). At the county level, 36–40 isolates were made from animals with between 3 and 7 isolates from milkers which was also not significantly different (χ2 = 1.989, *p* = 0.575). Random sampling stratified by dairy farm size (Livestock Unit or LU) also did not reveal significant associations between *S. aureus* isolates (human or animal) and LU classes (χ2 = 1.135, *p* = 0.769).

### 3.2. Microbiological Antibiotic Resistance and Susceptibility Testing (AST)

For the 170 isolates from cattle and humans, 25 antibiotics were considered relevant for humans as well as for animals as indicated in Table 2. By commercial preferences, 7/20 farms are using Mamifort (ampicillin) as a first choice in mastitis treatment, 8/20 farms are using Clamox LC (amoxicillin/clavulanate), and 5/20 farms are using the Mastijet (tetracycline). The outcomes detailing susceptible (S) and resistant (R) classifications along with MIC values from the Micro Scan Walk Away 40 SI are presented in Table 2. To determine whether methicillin-resistant *S. aureus* (MRSA) was present, screening for cefoxitin was carried out. The results showed that 30 (25%) of the bovine isolates were cefoxitin-resistant, the remaining 120 strains being susceptible. None of the human isolates were identified as MRSA. There was no significant difference in the frequency of MRSA in humans and cattle (χ2 *=* 4.857, *p* = 0.28). All isolates were resistant to ampicillin and penicillin.

A large proportion of bovine and human *S. aureus* isolates showed susceptibility to a variety of antimicrobials, including fosfomycin (86.67% of bovine strains and 100% of human strains), ciprofloxacin (86.67% and 90%, respectively), netilmicin (85.33% and 100%, respectively), levofloxacin (85.33% and 90%, respectively) and gentamicin (84% and 80%, respectively).

The frequency of phenotypic susceptibility (S) to *S. aureus* in the area studied was influenced by the source of the sample (human versus bovine strains χ2 = 35.726, *p* = 0.017), whether or not the strain was identified as MRSA (χ2 = 111.184, *p* < 0.000), county (χ2 = 97.331, *p* = 0.002), farm (χ2 = 681.778, *p* < 0.000) but not by the farm size (number of LUs) (χ2 = 69.996, *p* = 0.184). In the same way, the phenotypically resistant (R) *S. aureus* isolates were influenced by the origin of sample (human versus bovine χ2 = 36.510, *p* = 0.013), MRSA (χ2 = 108.891, *p* < 0.000), county (χ2 = 103.282, *p* < 0.000), farm (χ2 = 740.841, *p* < 0.000) but not by the LU of the farm (χ2 = 65.036, *p* = 0.306).

The MARI was calculated for each sample as a phenotypically resistant (R)/total antibiotics tested, for each of the 170 positive *S. aureus* isolates. At the level of the study, the mean and standard error of MARI was 0.590 ± 0.023, being higher for MRSA (0.898 ± 0.019) than non-MRSA (0.524 ± 0.024) isolates. In the area studied, neither the source of the sample (human or bovine t = 1.395, *p* = 0.092), county (F = 0.518, *p* = 0.671) or the LUs of the farm (F = 0.042, *p* = 0.988) were statistically associated with variations in the MARI. The MRSA (t = 8.010, *p* < 0.000) and farm influences (Kruskal–Wallis Test, χ2 = 45.758, *p* = 0.001) appeared to impact the MARI.

### 3.3. Prevalence of Resistance Genes

Table 3 presents the prevalence of resistance genes *blaZ*, *cfr*, *erm(B)*, *erm(C)*, *mecA*, and *tet(K)* (RG+ and SG+) across tested antibiotics, detected in susceptible (S) and resistant (R) isolates. It also includes the penetrance of these genes (P%) and diagnostic odds ratios (DOR) for positive phenotypic resistance in *S. aureus* from both cows and humans. Out of the 170 isolates tested for 13 antibiotics and harboring the *blaZ*, *cfr*, *erm(B)*, *erm(C)*, *mecA*, and *tet(K)* genes (RG+), 1510 out of 1946 (77.6%) exhibited the resistant phenotype (R) as identified by AST using Micro Scan Walk Away. In contrast, among isolates possessing the genes studied (SG+), 1070 out of 1794 (59.6%) showed the susceptible phenotype (S) in AST. The genes studied were present in 69% of isolates [(1510 + 1070)/(1946 + 1794)], calculated as the sum of RG+ and SG+ divided by the total number of resistant (R) and susceptible (S) values. Only 436 out of 1946 isolates (22.4%) did not possess the genes studied (RG−) and exhibited the resistant phenotype (R) in AST.

The study did not show an association between genetic resistance (G+ for the genes studied) and origin of the sample (χ2 = 9.331, *p* = 0.097), county (χ2 = 21.732, *p* = 0.115), but G+ was associated with the farm (χ2 = 173.559, *p* < 0.000), and the size of the farm (χ2 = 26.516, *p* = 0.033), in terms of LU index. For six genes studied in West Romania, the genotypic negative (G−) and positive (G+) class of *S. aureus* was not associated with the source of sample (χ2 = 9.331, *p* = 0.097) or county (χ2 = 21.732, *p* = 0.115), but it was influenced by MRSA (χ2 = 21.337, *p* = 0.001), farm (χ2 = 173.559, *p* < 0.000) and farm LU (χ2 = 26.516, *p* = 0.033).

### 3.4. Penetrance of the Resistance Genes

Presuming that resistance could be the result of other plasmid-borne genes or chromosomally encoded genes—factors not encountered in this study—in order to qualify the penetrance of the resistance genes, isolates characterized by phenotypic resistance, which did not carry the resistance genes, were not included in our evaluation. Table 3 displays the estimated penetrance (P%) of the genes based on AST interpretations.

The penetrance was 59% [P% = 1510/(1510 + 1070)] for all the genes and antibiotics tested in our area of the study. Regarding *blaZ*, the overall penetrance across the three antibiotics tested was 83%. Penetrance values specific to antibiotics associated with resistance conferred by the *blaZ* gene were as follows: 49% for amoxicillin/clavulanate, and 100% for ampicillin and penicillin (see Table 3). For the *mecA*, the penetrance for the antibiotics tested was 57%.The penetrance values were 100% for penicillin and ampicillin, 51–52% for amoxicillin/clavulanate andcefalotinand, and 20% for oxacillin (Table 3).In the case of *cfr*, *erm(B)* and *erm(C)*,the penetrance for the four antibiotics tested was around 50%. For the genes mentioned, the higher penetrance was in the case of clindamycin (69–82%—see Table 3). For *tet(K)*, the penetrance was 32%, indicating that one-third of *S. aureus* isolates exhibiting phenotypic resistance carried the *tet(K)* gene (Table 3).

Calculating the penetrance for each 1of the 170 *S. aureus* isolates, as PEN% = RG^+^/(RG^+^ + SG^+^), for all six genes tested, the average and standard error of PEN% was 0.361 ± 0.020, and higher in MRSA (0.667 ± 0.046) than non-MRSA (0.295 ± 0.019) isolates. In the area studied, neither the source of the sample (human or bovine, t = 0.312, *p* = 0.755), county (F = 0.518, *p* = 0.671) nor the LU of the farm (F = 0.043, *p* = 0.988) were associated with the variations of the PEN%, but the MRSA (t = 8.010, *p* < 0.000) and farm influences appeared to impact the index (Kruskal–Wallis Test, χ2 = 45.279, *p* = 0.001).

The PEN% for *S. aureus* appeared to be strongly positively correlated with MARI with a Pearson coefficient at *r* = +0.878 and *p* < 0.000. Also, the penetrance (PEN%) can be regressed against the multiple antibiotic resistance (MAR) index as a predicting variable, by the following relationship:PEN% value = −0.115 + (0.806 × MARI value), (F = 568.880, *p* < 0.000, R^2^ = 0.772)

The MARI significantly predicted PEN% which indicates that the MARI has interdependence on penetrance. Moreover, the R^2^ = 0.772 indicates that the regression explains 77.2% of the variance of PEN%.

### 3.5. AST Diagnostic Odds Ratio of Positive Phenotypic Resistance (DOR)

We have used the diagnostic odds ratio (DOR) of the positive phenotypic resistance (R+) in order to estimate the way in which phenotypic AST is marking the presence of resistance, as depicted in Table 3. For example, in the case of the *mecA* gene, we found examples such as amoxicillin/clavulanate (DOR = 1.18), cefoxitin (DOR = 2.08) and oxacillin (DOR = 1.94), for the *cfr* gene, chloramphenicol (DOR = 2.84), synercid (DOR = 3.27) and clindamycin (DOR = 2.75); for the *erm(B)* gene, clarithromycin (DOR = 1.66) and erythromycin (DOR = 1.19); for the *erm(C)* gene, clarithromycin (DOR = 2.00), erythromycin (DOR = 2.14) and synercid (DOR = 2.47); and for the *tet(K)* gene, tetracycline (DOR = 2.09). According to the data shown in Table 3, the study DOR index was 2.34 [DOR = (1510 × 724)/(436 × 1070)]; this means that the AST can be an acceptable estimate for the presence of genotypic resistance in the area of the study. On the other hand, we also found many examples where the DOR value was much lower than 1, for example the *blaZ* gene (DOR = 0.35).

## 4. Discussion

Bovine intramammary inflammation is a real problem for the dairy industry. Incountries where infectious/contagious mastitis remains a problem, the major pathogens are *S. aureus* and *S. agalactiae*. In human medicine, after the introduction of penicillin in the 1940s, resistance to the antibiotic developed very rapidly. As new antibiotics have been introduced, resistance has emerged with varying degrees of rapidity, such that many human bacterial strains are now multi-resistant. Similarly, the use of chemotherapy to treat bovine mastitis has led to multi-resistance. Where measures such as teat-dipping have not been introduced, multi-resistant *S. aureus* remains a problem. Identifying the antibiotic resistance profile of the bacteria causing infections is vital to ensuring effective chemotherapy [61]. *S. aureus* strains transfer readily from milking cattle to milkers/attendants and may be isolated from the human nasal cavity [62]. It is likely that transfer may also occur in the opposite direction. One limitation of the study is the absence of sequence type determination, which would have been useful for tracking the spread of specific clones. However, this did not impact the study’s primary focus on *S. aureus* resistance. Antibiotic susceptibility testing is an essential assay in developing a cautious and sensible approach towards the more rational use of antimicrobials for the treatment of bovine mastitis, as well as reducing overall use [63,64].

In the present study, *S. aureus* was found in 46.15% (n = 150) of milk samples and in 66.66% (n = 20) of samples from human attendants/milkers. The high prevalence is likely due to the poor hygiene practices in the farms during milking which were observed during the samplingvisits. It is interesting to compare data from similar European countries. A recent study in Italy found 47% (398 of 844 samples) of milk samples contained *S. aureus* [65]. The prevalence of *S. aureus* in cases of bovine mastitis in Germany was 21.8% (569 of 2614 samples) [66], while the comparable figure for the Czech Republic was much lower at 9% (60 of 669 samples) [67]. In the US, the prevalence in 189 dairy farms was much higher at 62.4%(118 of 189 herds) [68]. There is no indication as to how far this variation is the result of teat hygiene measures being introduced or not.

The antibiograms indicated that *S. aureus* exhibited the highest resistance rates to penicillin (100%) and ampicillin (100%), followed by clindamycin (74.67%), cefalotin (53.33%), and amoxicillin/clavulanic acid (52%). In one study from the Czech Republic, almost all tested antibiotics were susceptible. In 27.7% of isolates, resistance for ampicillin was recorded while for other antibiotics, the phenomenon was encountered for norfloxacin and tetracycline with a very low frequency [67]. In another study on *S. aureus* strains from Switzerland, similar levels of resistance were found to penicillin (39.2% of strains), and to ampicillin (26.7%), gentamicin (45.5%), and the highest levels of resistance werefound in oxacillin (64.7%) and lincomycin (53.7%) [58]. Relatively lower levels of resistance to penicillin (16.7%) and tetracycline (14.2%) were found in Greek isolates [69]. There seems little relationship between these patterns and antibiotic use. Differences in levels of hygiene and carry-over of centralized disease control in former Eastern European States might provide a clearer clue and might be investigated.

MRSA poses a significant challenge in both human and animal populations [70]. In our study, 20% of the cow isolates tested positive in cefoxitin screening, with the *mecA* gene detected in 26 isolates (86.67%), confirming MRSA presence [71]. Studies from Turkey, Greece, and Jordan reported MRSA prevalences of 75.4%, 81.3%, and 31.8%, respectively [72,73,74]. The primary contributors to the increased prevalence of MRSA are excessive and inappropriate use of beta-lactam antibiotics, along with poor hygienic conditions during milking [75]. The level of MRSA was lower in our study although it is clearly present in the national dairy herd. Not all farms accepted samples from milkers (see Figure 3) but, interestingly, MRSA was not found in milking staff which may possibly reflect a combination of the lower level of MRSA in milking cattle, and improvements in the extension service in Romania (in 3 of these 20 farms) with increased use of voluntary milking systems involving the mandatory use of disposable gloves.

This study focused on six genes for genetic detection of antibiotic resistance: *blaZ*, *cfr*, *erm(B)*, *erm(C)*, *mecA*, and *tet(K).* These were selected because they were the most appropriate antibiotics found in the panel [76] used for antibiograms. The genes tested in phenotypically resistant *S. aureus* isolates were all high, ranging from 57% (*cfr*) to 93% (*blaZ*). The prevalence of resistance genes in Poland has been tested 64% in the case of the *tet(K)* gene and 82% in the case of *blaZ* gene [77], but with no indication as to hygiene measures and other important contributory factors. In a recent study in the US, the prevalence for resistance genes was 4% for *blaZ*, 0.8% for *mecA*, 0.8% for *erm(B)* and 1.6% for *tet(K)* gene [68]. Our results are in contrast to the results from the US dairy sector [68] which experiences lower prevalences of MRSA and other multi-drug-resistant strains.

Clearly, amongst the classes of antibiotics, the penicillins and lincosamides should not be used for mastitis treatment in Romania and many other countries, as they are largely ineffective. This is certainly also true for penicillins for treatment of human infections. The antibiotic classes currently showing greatest efficacy in *S. aureus* infections are aminoglycosides, fluoroquinolones, phosphonic acid derivatives, steroid antibacterials, sulfonamide-trimethoprim combinations and second-generation cephalosporins. However, not all of these are in veterinary use (e.g., fluoroquinolones are used in human medicine and in pets, but not in farm animals). No new class of antibiotics has been introduced in veterinary medicine for the last 40 years. Therefore, in this context, we may suggest new tools for managing mastitis treatment with the available antibiotics under AMR phenomenon.

Utilizing the diagnostic odds ratio (DOR) of positive phenotypic resistance can effectively indicate the presence of resistance, particularly when the epidemiology of a specific geographic region is well characterized. In our study, the DOR was 1.85 and, generally, DOR values higher than 1 indicate that phenotypic resistance can be used to detect isolates possessing genes encoding resistance to antibiotics, with the advantage that no genetic analyses are required. However, our study revealed some DOR values lower than 1 for which there was no indication of the presence of relevant AMR genes, although a phenotypically susceptible phenotype may nevertheless contain AMR genes. This clearly requires further investigation.

The MARI was 0.590, being higher in MRSA than non-MRSA isolates. Even in non-MRSA isolates, the cut-off values were higher than reported elsewhere in the literature [78,79,80,81,82]. Bacteria having MARIs of ≥0.2 originate from a high-risk source of contamination, where several antibiotics are used [83,84]. The correlation of the MARI and PEN percentage means that in the area studied, phenotypic resistance is shown to be related to genotypic resistance and the MARI cut-off value may be used equally well for estimating the PEN% cut-off. In this study, for estimating the extent to which gene carriage corresponds to the phenotypic resistance defined as resistant or susceptible in AST, we again considered the notion of penetrance, which may be used as an indicator of multi-resistance comparable with the MARI with which it was positively strongly correlated (*r* = +0.878, *p* < 0.000). We have previously suggested including this indicator for evaluating the likely failure of an antibiotic treatment regime [85] at a possible cut-off ≤ 0.2, similar with the MARI index. For its calculation, we had two issues in view: (i) we did not consider resistant isolates negative for the genes, based on the presumption that resistance was generated through other mechanisms or plasmid-borne genes which were not included in this study, and (ii) we proceeded under the presumption that the emphasized resistance in resistant isolates testing positive of the genes was influenced by the genes. The penetrance for *blaZ* was 83%; for *mecA*,57%; for *cfr* 56%, for *erm(B)*, 50%; for *erm(C)*, 53%; and for *tet(K)*, 32%, all values being higher than the proposed 0.20 cut-off, and in concordance with poor clinical effect of the treatments. Using the regression equation in the studied samples, the PEN% value estimated by MARI (0.590) was 0.3605, very close to the value 0.361, as determined in this study. Under our regression estimation, MARI values under 0.4 also keep the PEN% values under 0.20. The dairy sector is a high-risk sector, where antibiotics are overused and, as a result the MARI is frequently higher than 0.20. According to our results, the MARI was 0.59 in the Western Romania dairy sector, as compared to other data—0.33 in Brazilian cows [83,86] and 0.52 in Egyptian cows [87]. In fact, PEN values over 0.20, as well as MARI values over 0.4, raise questions over the validity of current antimicrobial treatment programs or strategies, which suggests the need to alter these. Other future studies are recommended to calculate the clinical efficiency of mastitis treatments more precisely, as well as the association with the presented indicators such as PEN% and MARI.

## 5. Conclusions

This comparative study of phenotypic and genotypic AMR has highlighted the difficulties of tackling AMR *S. aureus* infection in the Romanian dairy industry. We have investigated the utility of penetrance (the percentage of strains exhibiting resistance as a phenotype compared to those carrying the resistance genotype) as a valuable parameter. Considering this value, along with the multiple antibiotic resistance index, we infer that penetrance could serve as a useful parameter for more precise targeting of chemotherapy for *S. aureus*. It highlights the relationship between phenotypic and genotypic resistance, particularly focusing on a specific set of resistance genes associated with antibiotics used in a particular farm or sector.

## Figures and Tables

**Figure 1 animals-14-02266-f001:**
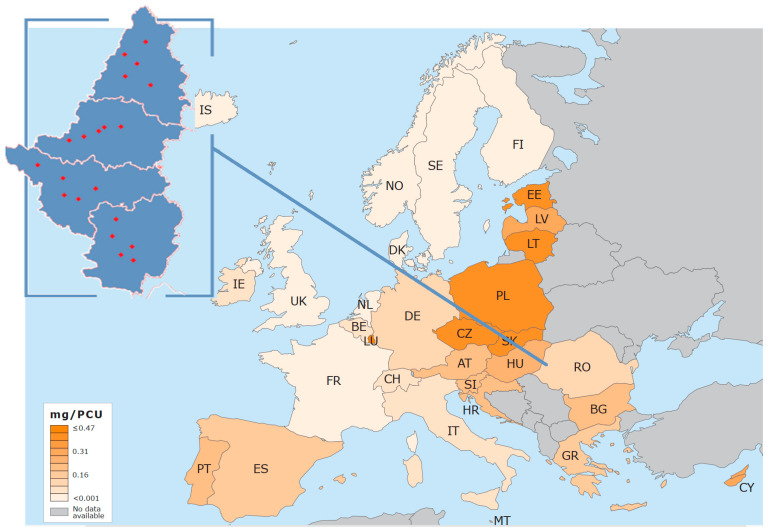
Geographical distribution of sales, in mg/PCU, of antibiotics for food-producing animals in 31 European countries in 2022, generated by the ESVAC database of European Medicine Agency [49], completed and adapted by authors and the geographical area of the study (left upper with farms marked as red spots).

**Figure 2 animals-14-02266-f002:**
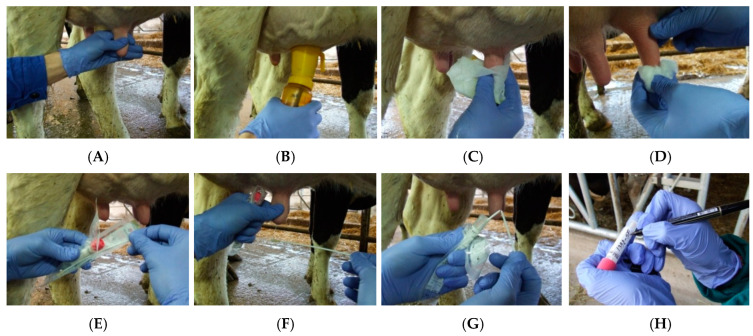
Preparation of the teat for milk sample collection for microbiological examination and sampling onto the swab of the collection and transport system in liquid medium—eSwab. (**A**)—Removal of the bacterial plug; (**B**)—disinfection of the teat; (**C**)—wiping off the disinfectant solution after 20 s; (**D**)—disinfection of the teat orifice; (**E**)—unsealing the collection swab; (**F**)—directing the milk stream towards the swab; (**G**)—shortening the shaft and inserting the swab into the liquid medium of the eSwab system; (**H**)—individualization and tight sealing of the tube with transport medium and swab.

**Figure 3 animals-14-02266-f003:**
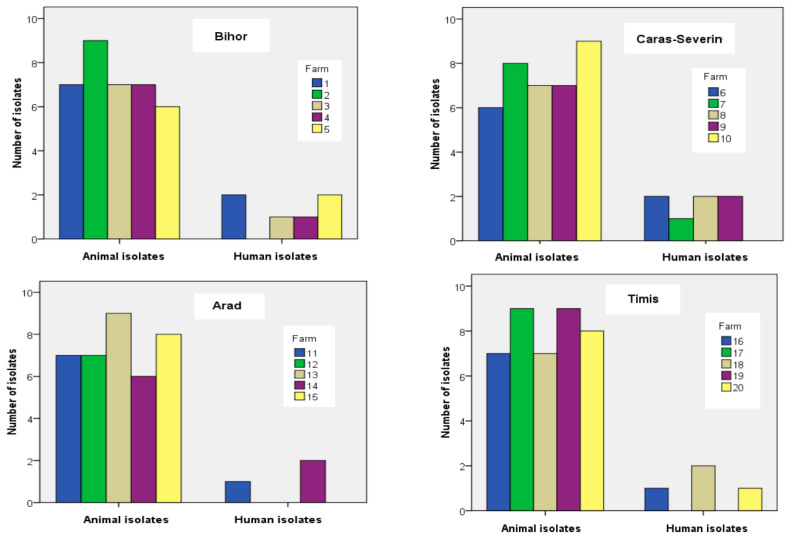
The numbers of animal and human isolates collected from the selected 20 farms from the four counties of West Romania.

**Table 1 animals-14-02266-t001:** Primers used for the amplification of resistance genes.

Gene	Primer	Primer Sequence	Annealing Temperature (°C)	Amplicon Size (bp)	Authors
		(5′-3′)			
*blaZ*	blaZFw	ACT TCA ACA CCT GCT GCT TTC	60 °C	490	Abdolmaleki Z et al., 2019 [50]
	blaZ R	TGA CCA CTT TTA TCA GCA ACC			
*cfr*	cfrFw	ATG AAT TTT AAT AAT AAA ACA AAG	58 °C	746	Kehrenberg et al., 2006 [51]
	cfr R	TAC ACC CAA AAT TAC ATC CG			
*erm(B)*	erm(B)Fw	CAT TTA ACG ACG AAA CTG GC	60 °C	745	Malhotra-Kumar et al., 2005 [52]
	erm(B) R	GGA ACA TCT GTG GTA TGG CG			
*erm(C)*	erm(C)Fw	ATC TTT GAA ATC GGC TCA GG	58 °C	299	Klare I. et al., 2007 [53]
	erm(C) R	CAA ACC CGT ATT CCA CGA TT			
*mecA*	mecAFw	CTG ATG GTA TGC AAC AAG TCG	55 °C	533	Lee, 2003 [54]
	mecA R	TGA GTT CTG CAG TAC CGG ATT			
*tet(K)*	tet(K)Fw	GTA GCG ACA ATA GGT AAT AGT	60 °C	360	Abdolmaleki et al., 2019 [50]
	tet(K) R	GTA GTG ACA ATA AAC CTC CTA			

**Table 2 animals-14-02266-t002:** Antibiotic susceptibility testing (AST) ^1^ results for *Staphylococcus aureus* (n = 170 isolates).

Class	Antibiotic	MIC (µg/mL)	Susceptible (S)		Resistant (R)		Total Isolates	
			Cows	Humans	Cows	Humans	S	R
Aminoglycosides	Gentamicin	4–8	126 (84%)	16 (80%)	24 (16%)	4 (20%)	142	28
	2. Netilmicin	8–16	128 (85.3%)	20 (100%)	22 (14.7%)	0 (0%)	148	22
Amphenicols	3. Chloramphenicol	8–16	106 (70.7%)	20 (100%)	44 (29.3%)	0 (0%)	126	44
Beta-lactam-lactamase-inhibitor (1st gen. Cephalosporins, 2nd gen. Cephalosporins and Penicillins)	4. Amoxicillin/ clavulanicacid	4/2	72 (48%)	12 (60%)	78 (52%)	8 (40%)	84	86
	5. Cefalotin	8–16	70 (46.7%)	12 (60%)	80 (53.3%)	8 (40%)	82	88
	6. Cefoxitin	4	120 (80%)	20 (100%)	30 (20%)	0 (0%)	140	30
	7. Ampicillin	0.25–8	0 (0%)	0 (0%)	150 (100%)	20 (100%)	0	170
	8. Oxacillin	4	119 (79%)	20 (100%)	31 (21%)	0 (0%)	139	31
	9. Penicillin	0.03–8	0 (0%)	0 (0%)	150 (100%)	20 (100%)	0	170
Fluoroquinolones	10. Ciprofloxacin	1–2	130 (86.7%)	18 (90%)	20 (13.3%)	2 (10%)	148	22
	11. Levofloxacin	1–4	128 (85.3%)	18 (90%)	22 (14.7%)	2 (10%)	146	24
	12. Moxifloxacin	0.5–1	120 (80%)	18 (90%)	30 (20%)	2 (10%)	138	32
Glycopeptides	13. Vancomycin	0.5–16	96 (64%)	16 (80%)	54 (36%)	4 (20%)	112	58
	14. Teicoplanin	1–16	104 (69.3%)	16 (80%)	46 (30.7%)	4 (20%)	120	50
Lincosamides	15. Clindamycin	0.25–2	38 (25.3%)	12 (60%)	112 (74.7%)	8 (40%)	50	120
Lipopeptides	16. Daptomycin	1.4	96 (64%)	16 (80%)	54 (36%)	4 (20%)	112	58
Macrolides	17. Clarithromycin	2–4	84 (56%)	10 (50%)	66 (44%)	10 (50%)	94	76
	18. Erythromycin	0.5–4	80 (53.3%)	12 (60%)	70 (46.7%)	8 (40%)	92	78
Oxazolidinones	19. Linezolid	4–8	147 (98%)	20 (100%)	3 (2%)	0 (0%)	167	3
Phosphonics	20. Fosfomycin	32	130 (86.7%)	20 (100%)	20 (13.3%)	0 (0%)	150	20
Rifamycins	21. Rifampicin	1–2	108 (72%)	16 (80%)	42 (28%)	4 (20%)	124	46
Steroid antibacterials	22. Fusidic-acid	2.16	120 (80%)	16 (80%)	30 (20%)	4 (20%)	136	34
Sulfonamide-trimethoprim—combinations	23. Sulfamethoxazole/trimethoprim	2/38	120 (80%)	18 (90%)	30 (20%)	2 (10%)	138	32
Streptogramins	24. Synercid	1–2	94 (62.7%)	16 (80%)	56 (37.3%)	4 (20%)	110	60
Tetracyclines	25. Tetracycline	2–8	106 (70.7%)	16 (80%)	44 (29.3%)	4 (20%)	122	48

^1^ Processed outputs of the MicroScan Walk Away System for identified *Staphylococcus aureus* isolates (150 isolates from mastitis and 20 from human samples).

**Table 3 animals-14-02266-t003:** The distribution of resistance genes among resistant and susceptible *Staphylococcus aureus* strains (n = 170 isolates), along with gene penetrance (P%) and diagnostic odds ratios (DOR).

Genes	Antibiotics	R			S			Penetrance (%)	DOR
		Total	RG+	RG−	Total	SG+	SG−		
*blaZ*	Amoxycillin/clavulanicacid	86	78(91%)	8(9%)	84	82(98%)	2(2%)	49%	0.24
	Penicillin	170	160(94%)	10(6%)	0	0	0	100%	-
	Ampicillin	170	160(94%)	10(6%)	0	0	0	100%	-
	Subtotal	426	398	28	84	82	2	83%	0.35
*mecA*	Amoxycillin/clavulanic acid	86	68(79%)	18(21%)	84	64(76%)	20(24%)	51%	1.18
	Ampicillin	170	132(78%)	38(22%)	0	0	0	100%	-
	Cefalotin	88	68(77%)	20(23%)	82	64(78%)	18(22%)	52%	0.96
	Cefoxitin screening	30	26(87%)	4(13%)	140	106(76%)	34(24%)	20%	2.08
	Oxacillin	31	27(87%)	4 (13%)	139	108(78%)	31(22%)	20%	1.94
	Penicillin	170	132(78%)	38(22%)	0	0	0	100%	-
	Subtotal	575	453	122	445	342	103	57%	1.12
*cfr*	Chloramphenicol	44	28(64%)	16(36%)	126	48(38%)	78(62%)	37%	2.84
	Linezolid	3	3(100%)	0(0%)	167	4(2%)	163(98%)	43%	-
	Synercid	60	38(63%)	22(37%)	110	38(35%)	72(65%)	50%	3.27
	Clindamycin	120	62(52%)	58(48%)	50	14(28%)	36(72%)	82%	2.75
	Subtotal	227	131	96	453	104	349	56%	4.58
*erm(B)*	Clarithromycin	76	58(76%)	18(24%)	94	62(66%)	32(34%)	48%	1.66
	Clindamycin	120	84(70%)	36(30%)	50	36(72%)	14(28%)	70%	0.91
	Erythromycin	80	58(70%)	22(30%)	90	62(69%)	28(31%)	48%	1.19
	Synercid	60	42(70%)	18(30%)	110	78(71%)	32(29%)	35%	0.96
	Subtotal	336	242	94	344	238	106	50%	1.15
*erm(C)*	Clarithromycin	76	58(76%)	18(24%)	94	58(62%)	36(38%)	50%	2
	Clindamycin	120	80(67%)	40(33%)	50	36(72%)	14(28%)	69%	0.78
	Erythromycin	78	60(77%)	18(23%)	92	56(61%)	36(39%)	52%	2.14
	Synercid	60	48(80%)	12(20%)	110	68(62%)	42(38%)	41%	2.47
	Subtotal	334	246	88	346	218	128	53%	1.64
*tet(K)*	Tetracycline	48	40(83%)	8(17%)	122	86(70%)	36(30%)	32%	2.09
	Subtotal	48	40	8	122	86	36	32%	2.09
Total study		1946	1510	436	1794	1070	724	58.53%	2.34

RG+: Refers to phenotypically resistant isolates that possess the resistance gene. RG−: refers to phenotypically resistant isolates that do not possess the resistance gene. SG+: refers to phenotypically susceptible isolates that possess the resistance gene. SG−: refers to phenotypically susceptible isolates that do not possess the resistance gene. DOR: diagnostic odds ratio.

## Data Availability

All of the data presented were obtained from subjects involved in this study. The names of companies and farms are under a confidentiality agreement.

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
