# Peer review of "Microbiological and Molecular Investigation of Antimicrobial Resistance in Staphylococcus aureus Isolates from Western Romanian Dairy Farms: An Epidemiological Approach"

_animals, 2024, doi:10.3390/ani14152266_

Round 1

Reviewer 1 Report

Comments and Suggestions for Authors

Dear authors, 

I really appreciate your manuscript. It is well written and rich in scientific contents. I highlithed some minor comments and correction. The main limitation I found is the lack of "real" genetic characterization; you work only  on genes without performing MLST analysis or whole genome sequencing. However, I hope you will provide these interesting results in a new paper!

Comments on the Quality of English Language

Minor 

Author Response

Dear Reviewer,

We would like to thank you for your review, suggestions, and comments. We have made modifications, corrections, and added more information to make our paper clearer and more understandable.

Please see our revised paper (Revision 1).

Thank you again!

Ioan H

Reviewer 2 Report

Comments and Suggestions for Authors

The article examines the relationship between antibiotic resistance genes and the phenotypic expression of antibiotic resistance and susceptibility in cows with mastitis in western Romania. In general, the authors discuss in some detail the problem of antimicrobial therapy in the dairy industry and identify possible ways to solve it. I have only minor comments

The title of the article should indicate the region from where the samples were taken. Perhaps you should write "... isolates from the dairy farms of Western Romania ..."

Section 2.1 Were cows with clinical or subclinical mastitis included in the study?

Section 2.1 Please provide more information about the cows. In particular, breed, age, number of milking days, number of calves, milk yield, etc. 

Section 2.1 Describe the process of selection and transportation of samples in more detail.

Line 174. Why were so many cycles selected for PCR?

Section 2.4. Please indicate how many replicates you used when performing PCR.

Line 224. It is not very clear for what type of analysis this or that statistical test was used.

Figure 2 and Table 2 should be transferred to those sections that describe the results reflected in this figure and table

Lines 356-369. The first paragraph of the Discussion is a complete repetition of the Introduction. This part should be removed or shortened to 1-2 sentences

Line 371-373. Why is this evident?

Line 373-376. If you mention these articles, please indicate the number of samples in these studies. Because the meanings of the mentioned percentages are not very clear

Author Response

Dear Reviewer,

We would like to thank you for your review and comments. Please find our responses and the suggested modifications incorporated into the paper.

Thank you!

Reviewer 3 Report

Comments and Suggestions for Authors

The manuscript titled "Characterization of antimicrobial resistance in Staphylococcus aureus isolates from dairy farms – an epidemiological approach for more precise targeting of chemotherapy" provides an insightful and comprehensive analysis of the antimicrobial resistance (AMR) patterns in Staphylococcus aureus isolated from dairy farms in Western Romania. The research offers valuable data on the prevalence of AMR in both bovine and human samples, focusing on methicillin-resistant Staphylococcus aureus (MRSA) and the distribution of resistance genes, which is crucial for developing targeted antimicrobial strategies.

The methodology, including random sampling from 20 farms across four counties, provides a reliable dataset. The inclusion of both bovine and human samples allows for a comparative analysis that highlights the zoonotic potential of S. aureus. The use of phenotypic and genotypic methods, including q-PCR for detecting resistance genes, enriches the study's findings. This dual approach ensures a more accurate representation of the AMR landscape. The study presents a thorough AST, covering a wide range of antibiotics, which is crucial for understanding the resistance profile of S. aureus isolates. The clear differentiation between MRSA and non-MRSA isolates provides nuanced insights into the severity of AMR. The study offers valuable epidemiological data such as the prevalence of specific resistance genes and their penetrance. This information is essential for designing effective antimicrobial management strategies to the specific conditions in the region. The application of various statistical analyses, including correlation and regression, adds depth to the study, enhancing the reliability and interpretability of the results.

However, the study could benefit from a more detailed discussion of potential confounding factors, such as farm management practices, antibiotic usage history, and environmental conditions, which may influence the prevalence and distribution of AMR genes. Addressing these areas for improvement would further enhance the study's impact and practical applicability.

Author Response

Dear Reviewer,

We would like to thank you for your appreciation. We have uploaded a new revision to incorporate the modifications, suggestions, and comments from your colleagues (Reviewers 1 and 2).

We have made adjustments, corrections, and added more information to make our paper clearer and more understandable.

Please see our revised paper (Revision 1).

Thank you again!

Ioan H